# A High-Grade Lachman’s Exam Predicts a Ramp Tear of the Medial Meniscus in Patients with Anterior Cruciate Ligament Tear: A Prospective Clinical and Radiological Evaluation

**DOI:** 10.3390/jcm13030683

**Published:** 2024-01-24

**Authors:** Filippo Familiari, Luke V. Tollefson, Antonio Izzo, Michele Mercurio, Robert F. LaPrade, Giovanni Di Vico

**Affiliations:** 1Department of Orthopaedic and Trauma Surgery, Magna Graecia University, 88100 Catanzaro, Italy; filippofamiliari@unicz.it; 2Research Center on Musculoskeletal Health, MusculoSkeletal Health@UMG, Magna Graecia University, 88100 Catanzaro, Italy; 3Twin Cities Orthopedics, Edina, MN 55435, USA; lukevtollefson@gmail.com (L.V.T.); izzoantonio1992@gmail.com (A.I.); drlaprade@srpivail.org (R.F.L.); 4Department of Public Health, Trauma and Orthopaedics, University Federico II, 80138 Napoli, Italy; 5Casa di Cura San Michele, 81024 Maddaloni, Italy; giovannidivico@libero.it

**Keywords:** anterior cruciate ligament, medial meniscus ramp, Lachman’s exam, posterior tibial slope, rotatory instability, clinical diagnostics

## Abstract

**Background:** Medial meniscus ramp tears are present in 22.9–40.8% of anterior cruciate ligament tears. The diagnosis of ramp tears is difficult on MRI, with sensitivity reported around 48%, which has recently emphasized the importance of proper arthroscopic probing for ramp tears. **Methods:** A prospective evaluation was performed on patients undergoing a single bundle ACL reconstruction to assess patient demographics, posterior tibial slope, posterior cruciate ligament angle, Lachman’s exam, and rotational instability to determine secondary findings associated with medial meniscal ramp tears. **Results:** A total of 96 patients underwent ACL reconstruction, of these, 63 patients had an isolated ACL tear, and 33 patients had an ACL tear with a concomitant medial meniscus ramp tear. A high-grade Lachman’s exam and male sex were associated with medial meniscus ramp tears. There were no differences in posterior tibial slope, posterior cruciate ligament angle, or rotational instability between groups. **Conclusions:** This study found that a high-grade Lachman’s exam and male sex were significantly associated with patients with an ACL tear with a concomitant medial meniscus ramp tear. These findings suggest that an ACL tear with concomitant medial meniscus ramp tear may be better diagnosed based upon the clinical evaluation rather than other secondary radiological findings.

## 1. Introduction

Medial meniscus ramp tears are meniscal tears associated with anterior cruciate ligament (ACL) tears and have been reported to be more prevalent when an ACL reconstruction (ACLR) is delayed [1]. Medial meniscus ramp tears have been described as longitudinal tears of the posterior horn of the medial meniscus at the meniscocapsular junction, affecting either the meniscocapsular (superior attachment), meniscotibial (inferior attachment), or both attachments [1,2]. A complete separation of the meniscocapsular junction occurs when both the meniscocapsular and meniscotibial attachments are torn. Medial meniscus ramp tears are often called “hidden lesions” due to the difficulty in identifying the tear with standard anterior arthroscopic portals and are often diagnosed when the arthroscope is viewing the posteromedial compartment [3]. Arthroscopic probing can help to determine the stability of a medial meniscus ramp tear and a rasp should be used to properly identify the margins of the tear [3]. Proper identification is key in determining the extent of the injury and for the determination of the repair technique that should be utilized. As the recognition and awareness of ramp tears increases, their reported incidence is also increasing [3,4]; the incidence of ACL tears with medial meniscus ramp tears has recently been reported at 22.9–40.8% [5,6,7]. These incidence percentages have been increasing due to the increased knowledge about the prevalence and proper probing and repair techniques for medial meniscus ramp tears. MRI diagnosis of ramp tears is often difficult and not always reliable due to its low sensitivity, which emphasizes the need for proper arthroscopic evaluation [3,4,8].

Medial meniscus ramp tears have detrimental effects on overall knee stability and if left untreated, can contribute to the early progression of osteoarthritis [9]. Biomechanical studies have reported that concomitant ACL and medial meniscus ramp tears increase anterior tibial translation and internal rotation of the knee compared to an isolated ACL tear state [1,10,11]. Additionally, repairing a ramp tear can restore near native knee biomechanics compared to the unrepaired state [10,11]. In the clinical setting, it has been reported that ramp tears are associated with higher grades on the Lachman and pivot shift exams [12,13]. Additionally, studies have reported improved postoperative patient reported outcomes after undergoing ACLR with a medial meniscus ramp repair when compared to patients undergoing isolated ACLR [12,14,15]. Medial meniscus ramp tears are important to repair concomitant with an ACLR to properly restore knee biomechanics and prevent the early progression of osteoarthritis.

Medial meniscus ramp tears are commonly repaired using either an all-inside or an inside-out technique [16]. All-inside techniques can be performed with either an all-inside device or an all-inside suture hook [17]. Inside-out repair is more robust than an all-inside technique but requires an additional posteromedial portal to catch the sutures that exit the capsule [16,18]. Both techniques report improved patient reported outcomes and surgical outcomes. Typically, for larger tears, an inside-out repair offers a stronger fixation of the meniscus while an all-inside fixation is more suitable for smaller repairs. It is essential that proper arthroscopic probing is performed during assessment of the tear as more unstable medial meniscus ramp tears may require an inside-out repair instead of an all-inside repair [3]. Repair of the ramp tear in addition to the ACLR is critical to ensure that the near native biomechanical state of the knee is restored.

Various studies have attempted to improve the diagnostic capabilities for assessing medial meniscus ramp tears. Certain factors, including contact sports, pivot shift type bone bruises, and Segond fractures have all been mentioned as potential predictors of concomitant ACL and medial meniscus ramp tears [5,19]. A study reporting on the medial posterior tibial slope reported that increased medial posterior tibial slope, measured on MRI and only for patients with noncontact ACL injuries, were correlated to an ACL tear with a concomitant medial meniscus root tear [20]. The purpose of this study was to determine whether any other preoperative diagnostic tools can help to predict the presence of a medial meniscus ramp tear. The variables assessed include posterior tibial slope, posterior cruciate ligament (PCL) buckling angle, clinical exams, and patient demographics and were analyzed to determine if any correlation to ACL tears with concomitant medial meniscus ramp tears was present. Our null hypothesis was that there would be no significant differences between the cohorts of isolated ACLR and ACLR with ramp lesions.

## 2. Materials and Methods

The local institutional review board approved the study protocol and the research was conducted in compliance with the Declaration of Helsinki. Patients undergoing primary unilateral arthroscopic anatomic single-bundle ACL reconstruction between July 2016 and September 2020 from a single center were prospectively enrolled as the study group and informed consent was obtained.

### 2.1. Patient Involvement Statement

Eligibility criteria were as follows: (1) ≥18 years old at the time of surgery, (2) an ACL tear undergoing an ACLR (using semitendinosus and gracilis tendon autograft (ST-G)), and (3) the capability to communicate with health care professionals and provided an informed consent.

Exclusion criteria included previous surgeries of the affected knee, previous or concomitant lesions of the PCL and of the medial and lateral collateral ligaments (MCL and LCL), previous fractures of the ipsilateral femur and tibia, as well as patients showing severe knee osteoarthritis (Kellgren-Lawrence grade III and IV) [21,22].

The diagnosis was based on clinical and radiological evaluation of the knee (plain radiographs and MRI). A total of 103 patients met the study criteria. Demographic data, including age, gender, dominant limb defined as the preferred kicking limb, and the injury to surgery time interval were assessed [23].

The clinical assessment of patients included performing the Lachman and pivot shift exams. The degree of anterior–posterior translation was recorded with the Lachman exam and graded with Grade 1 (mild) for 3–5 mm translation of the tibia on the femur, Grade 2 (moderate) for 5–10 mm translation of the tibia on the femur and Grade 3 (severe) for >10 mm translation of the tibia on the femur. Rotatory instability was evaluated with the pivot shift test. It was graded with grade 1 (glide), grade 2 (anterolateral subluxation) and grade 3 (explosive pivot shift test).

### 2.2. Surgical Technique

All surgical procedures were performed by a single surgeon with a high level of experience in knee arthroscopy. Meniscal ramp tears were identified by exploration of the posteromedial compartment and repaired by an all-inside hook suture from an accessory posteromedial portal.

For the tibial tunnel for the ACLR, the aimer was adjusted to the 55° position to ensure adequate tibial tunnel length, and the guide tip was positioned intra-articularly through the anteromedial portal. Femoral tunnel drilling was performed via the anteromedial portal technique with the knee flexed to 120°; tunnel positions were placed within the native ACL footprints. The ST-G were harvested for a 4-strand graft aiming for a minimum graft diameter of 8 mm. In all ACLR procedures, a cortical suspension device was used for femoral fixation. A bioabsorbable interference screw was used for tibial fixation while the graft was tensioned.

### 2.3. Postoperative Rehabilitation

Postoperatively, patients ambulated with crutches, bearing weight as tolerated, and the range of knee motion was limited to 90° for one week. After one week, patients began a range of motion therapy. After nearly full range of motion was achieved, patients started strength training, with an emphasis on closed kinetic chain exercises. The preservation of quadriceps function in the early postoperative stage of rehabilitation after surgery was emphasized with an early initiation of isometric quadriceps setting exercises. Activities that prepared the individual for progression to full weight bearing and ambulation, improving balance and postural control, and achieving normal gait were allowed after 4 weeks. Weight-bearing strengthening activities were initiated and progressed after 6 weeks. Strengthening exercises through a full range of motion and activities to enhance neuromuscular control were also advanced to ensure full recovery and maintenance of strength and dynamic stability. Rehabilitation progressed from functional activities to sports-specific training. When the quadriceps limb symmetry index was 85% or greater and strength, proprioception, and endurance for functional progression were satisfied, the patients began full-effort sprinting, cutting, and plyometric activities.

### 2.4. Radiological Assessment

Radiographs and MRI scans of the involved knee were available for all patients evaluated. All imaging measurements were carried out on digital X-rays using a DICOM medical image viewer (Horos Project; Purview, Annapolis, MD, USA). On the standing lateral radiographic view, the posterior tibial slope was measured according to the method described by Dejour et al. [24]. This measurement was obtained by drawing a line between the center of two points, one just below the tibial tubercle and one 10 cm below that. Then a line is drawn from the most superior points of the anterior and posterior portions of the medial tibial plateau. The angle between these two lines is the posterior tibial slope (Figure 1). On the mid-sagittal view of the MRI, the angle formed between the proximal and distal portions of the PCL was measured according to the method described by Yoon et al. [25]. The PCL buckling angle is the angle between lines drawn in the central portions of the PCL femoral and tibial insertions (Figure 2).

All measurements were performed by two independent surgeons blinded to the aim of the study to assess the interobserver reliability. The same surgeons performed the same measurements after 4 weeks to assess the intra-observer reliability. During arthroscopy, associated meniscal tears were assessed and recorded.

### 2.5. Statistical Analysis

All data were collected, measured, and reported at an accuracy of one decimal place. Continuous variables are presented as the mean, standard deviation (SD), and range, and categorical variables are presented as counts. The distribution of the numeric samples was assessed using the Kolmogorov—Smirnov normality test. Based on this preliminary analysis, parametric tests were adopted. Correlations were tested to investigate possible associations among the available data, and Pearson’s coefficient or Phi’s coefficient was adopted when appropriate. The correlation was considered to be strong (r > 0.5), medium (0.5 < r < 0.3), or small (0.3 < r < 0.1). Uni- and multivariate linear regression was performed on the entire patient population to test for possible meniscal ramp tear predictors.

The explanatory and confounding pre- and postoperative variables included in the analysis were sex (categorical), age (continuous), dominant limb (categorical), injury to surgery interval (continuous), Lachman test grade (continuous), rotatory instability grade (categorical), posterior tibial slope (continuous), PCL buckling angle (continuous), and concomitant lesion (categorical). Only explanatory and confounding variables that showed a trend toward an association (e.g., *p* < 0.10) with the outcome of interest in the univariate analysis were included in the multiple regression analysis.

Post hoc power was calculated by considering the sample size, the observed effect size, and an α-value of 0.05; a post hoc power greater than 80% was considered appropriate. IBM SPSS Statistics software (version 26, IBM Corp., Armonk, NY, USA) and G*Power (version 3.1.9.2, Institut für Experimentelle Psychologie, Heinrich Heine Universität, Düsseldorf, Germany) were used to construct the database and perform statistical analysis. A *p*-value of less than 0.05 was considered significant.

## 3. Results

The final patient cohort consisted of 96 patients. Eighty-four (87.5%) of the patients were male and the patients had an average age of 28.6 ± 7.8 years (range, 18–49 years) at the time of surgery. The average time to surgery from injury was 9.9 ± 13.7 months (range, 0.8–60 months. The Lachman exam and rotatory instability (using pivot shift exam) findings, posterior tibial slope, PCL buckling angle, concomitant meniscal injuries, and time to follow-up were also recorded. The demographic and radiological characteristics of the included patients are summarized in Table 1.

A concomitant ACL tear and medial meniscus ramp tear were present in 33 (34.4%) patients. Overall, a concomitant tear of the anterior horn of the medial meniscus (outside the zone of a meniscus ramp tear area) and of the lateral meniscus was present in 15 (15.6%) and 18 (18.8%) patients, respectively.

The cohorts of patients with isolated ACL tears and ACL tears with concomitant medial meniscus ramp tears were separated. All demographic and radiologic factors were compared between groups to assess for associations (Table 2). A higher rate of male patients was found in the ACL tear with concomitant medial meniscus ramp tear group (*p* = 0.007). No significant differences were found in terms of age at the time of surgery (*p* = 0.722), injury to surgery interval (*p* = 0.255), rotatory instability (Grade 1, *p* = 0.645; Grade 2, *p* = 0.479; Grade 3, *p* = 0.689), posterior tibial slope (*p* = 0.192), PCL buckling angle (*p* = 0.495) or concomitant meniscal injuries (Anterior horn medial meniscus, *p* = 0.249; Lateral meniscus, *p* = 0.784) between the groups.

Cohen’s kappa coefficients for intraobserver and interobserver reliability of radiological assessments were 0.88 and 0.86 for MRI, and 0.86 and 0.87 for radiographs, respectively.

In the univariate regression analysis, a higher grade of the Lachman test was associated with the diagnosis of ACL tear with a concomitant medial meniscus ramp tear (*p* = 0.027, β = 0.216). Male sex was associated with the diagnosis of ACL tear with a concomitant medial meniscus ramp tear (*p* = 0.007, β = −0.191).

The multivariate analysis confirmed the association between the diagnosis of an ACL tear with a concomitant medial meniscus ramp tear with male sex (*p* = 0.013, β = −0.359) and a higher grade of the Lachman test (*p* = 0.049, β = 0.203).

## 4. Discussion

The most important finding from this study was that a high-grade Lachman exam was associated with an anterior cruciate ligament (ACL) tear and concomitant medial meniscus ramp tears compared to an isolated ACL tear. Overall, there were no significant differences in the posterior tibial slope, posterior cruciate ligament (PCL) buckling angle, or rotatory instability between the isolated ACL tear group and the ACL tear with medial meniscus ramp tear group. These findings suggest that medial meniscus ramp tears that occur with ACL tears may not be able to be diagnosed on the secondary sign of a difference of the posterior tibial slope angle or the PCL buckling angle, but rather are better diagnosed on the clinical examination with a higher-grade Lachman exam.

The finding from this study suggests that if a patient has a high-grade Lachman test, they are more likely to have an ACL tear with a concomitant medial meniscus ramp tear. This finding is consistent with the previous literature including a study by DePhillipo et al. [12], which reported that patients with an ACL tear and a medial meniscus ramp tear had significantly increased high-grade Lachman exams compared to patients with isolated ACL tears. They reported that 44% of patients with an ACL tear and medial meniscus ramp tear had a high-grade Lachman exam compared to 6% of patients with an isolated ACL tear. Additionally, biomechanical studies by Peltier et al. [26] and DePhillipo et al. [10] reported that medial meniscus ramp tears in addition to ACL tears increase anterior tibial translation compared to an isolated ACL tear state. Further studies reported that repair of these medial meniscus ramp lesion significantly improve knee biomechanics compared to the medial meniscus ramp tear state, but may not always return the knee to the near native state [10,11]. The meniscus ramp attachment at the posterior aspect of the medial meniscus is important for preventing increased anterior tibial translation and high-grade Lachman exams. In conjunction with our findings, the previous literature supports that a finding of a high-grade Lachman exam during preoperative clinical assessment can be associated with an ACL tear with a medial meniscus ramp tear and the probable presence of a ramp tear should be assessed for during arthroscopic evaluation.

Another interesting finding from this study was that the incidence of ACL tears with concomitant medial meniscus ramp tears was 34.4% and there was a significant association between male sex and an ACL tear with a concomitant medial meniscus ramp tear. All of the patients in the ACL tear with concomitant medial meniscus ramp tear cohort were males. Due to an increase in recognition of ramp tears, the recent literature has reported the incidence of ACL tears with medial meniscus ramp tears at 22.9–40.8% [5,6,7]. Findings from our study are in line with the previous literature with roughly a third of patients with an ACL tear also having a medial meniscus ramp tear. Recognition of medial meniscus ramp tears is notoriously difficult on both MRI and arthroscopic probing. A study by DePhillipo et al. [8], reported a sensitivity of 48% on MRI for diagnosing medial meniscus ramp tears, which emphasizes the need for proper arthroscopic identification and evaluation of ramp tears. Additionally, the finding of the association of the male sex as a risk factors for medial meniscus ramp tears is consistent with a systematic review by Kunze et al. [27]. These findings highlight the need for additional diagnostic tools to identify medial meniscus ramp tears. When certain indicators, like a high-grade Lachman exam and male sex are present, there is an increased probability that a medial meniscus ramp tear is present as well. In these cases, there should be an increased effort to verify the presence or absence of a medial meniscus ramp tear during arthroscopic probing. A high prevalence is being reported in the literature and the recognition and proper evaluation techniques are increasing.

In this study, there was no significant difference between posterior tibial slope, rotatory instability, and PCL buckling angle. Posterior tibial slope was noted to be lower in the ACL and ramp tear cohort; however, it was not significantly different from the isolated ACL group. A study by Gali et al. [28], reported that an increase in PCL buckling angle is associated with ACL tears suggesting that more anterior tibial translation is present when the ACL is torn. In our study, the findings suggest that PCL buckling angle may be associated with ACL tears but may not be associated with a concomitant medial meniscus ramp tear. Posterior tibial slope is a common diagnostic tool while assessing for ACL tears and is commonly referenced as a risk factor for ACL graft failure [29]. Increases in medial posterior tibial slope have been associated with increased incidence of medial meniscus ramp tears with ACL tears in the previously reported literature [30,31,32]. Furthermore, our findings do not report a significant association between a high-grade pivot shift exam and the presence of a medial meniscus ramp tear with an ACL tear. A study by Mouton et al. [13], reported that between patients with an ACL tear and a medial meniscus ramp tear or no medial meniscus ramp tear, there was no significant difference in rotatory instability using the pivot shift exam. Our findings suggest that medial meniscus ramp tears with ACL tears may not be diagnosed on the secondary sign of posterior PCL buckling angle, posterior tibial slope, or rotatory instability.

Repair of the medial meniscus ramp tears in conjunction with ACLR is important to improve patient outcomes. A study by Ahn et al. [14], reported a clinical success rate of 96.4% for patients undergoing either a modified all-inside or inside-out repair technique at the time of ACL reconstruction. These ramp repairs were assessed using second-look arthroscopy and patients with full healing or partial healing and no clinical symptoms were considered clinical successes, while patients with no healing or partial healing with clinical symptoms were considered clinical failures. A systematic review by D’Ambrosi et al. [15], reported that not all medial meniscus ramp lesions need to be repaired, however, all unstable medial meniscus ramp tears should be repaired. These findings again emphasize the importance of proper identification, probing, and rasping of ramp tears to determine the full extent and stability of the tear prior to repair. If a medial meniscus ramp tear in the presence of an ACL tear is unstable, it should be repaired to improve patient outcomes.

Concomitant tears to the medial and lateral menisci (excluding medial meniscus ramp tears) were found to be not significant between patients with ACL tears and ACL tears with concomitant medial meniscus ramp tears. Certain injury presentations including the “unhappy/terrible triad” have been presented to report on injuries of the ACL with other concomitant injuries including the menisci, the medial collateral ligament (MCL), and the anterolateral complex [7,33]. In this study, we did not find that the incidence of an ACL tear with a medial meniscus ramp tear was correlated with other meniscal pathology. A study by Magosch et al. [7], reported that meniscal injuries are present in 67% of ACL injuries, however, there was no specific combination of meniscal injuries that were present in a higher percentage, apart from 6% on ACL tears with both medial meniscus ramp tears and lateral meniscus root tears. Medial meniscus ramp tears are the most common pathology to occur with ACL tears, while lateral meniscus root tears are the second most common meniscal pathology occurring in 6.6–15% [34,35] of patients with ACL tears. Although concomitant ligamentous injuries do occur with ACL tears, this study did not attempt to correlate ACL tears with other ligamentous injuries like the MCL or posterolateral corner (fibular collateral ligament, popliteus tendon, and popliteofibular ligament).

This study is not without potential limitations. One potential limitation was the limited sample size, a larger sample size would help increase statistical power. Another limitation was that most of the patients with ACL tears in this study were males, so further studies to assess secondary signs of meniscal ramp tears in females would be indicated. Furthermore, other meniscus pathology that was present could have influenced the results presented. Future research around this topic is indicated as the literature on this subject is limited. Future studies should utilize larger patient cohorts and continue to report on posterior tibial slope, anterior tibial translation, Lachman’s exam, and the pivot shift exam to assess for medial meniscus ramp tears as clinical and biomechanical studies have reported that these diagnostic tools could help in assessing for ramp tears. Additionally, a cohort of patients with only an ACL tear and medial meniscus ramp tear should be compared to patients with a true isolated ACL to reduce confounding variables.

## 5. Conclusions

In conclusion, this study found that a high-grade Lachman’s exam and male sex were significantly associated with patients with an ACL tear with a concomitant medial meniscus ramp tear compared to patients with an isolated ACL tear. Posterior tibial slope, rotatory instability measured with the pivot shift exam, and PCL buckling angle were similar between cohorts. These findings suggest that ACL tear with concomitant medial meniscus ramp tear may be better diagnosed with clinical evaluation. When patients present with an ACL tear with a high-grade Lachman exam, a medial meniscus ramp tear should be suspected, and proper arthroscopic probing and diagnosis should be performed.

## Figures and Tables

**Figure 1 jcm-13-00683-f001:**
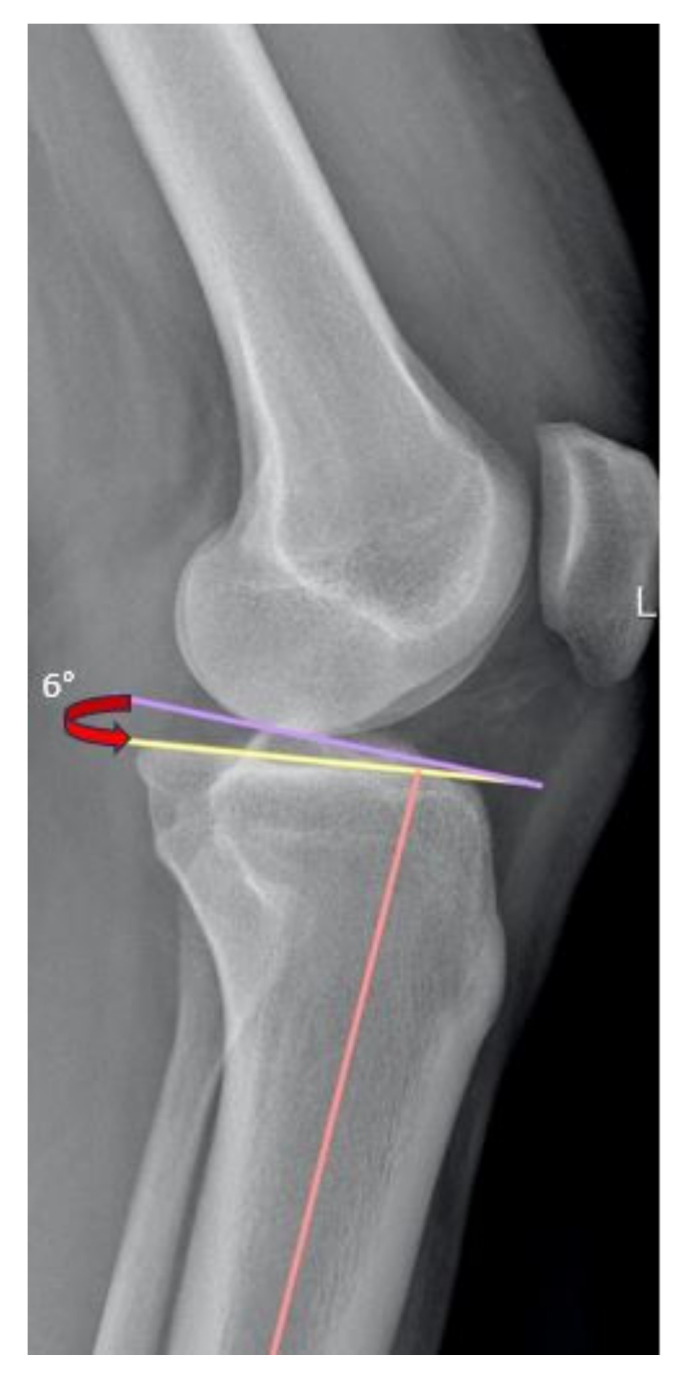
The posterior tibial slope was measured according to the method described by Dejour et al. [24] using the true sagittal view to measure the angle between the line perpendicular to the tibial diaphyseal axis (violet line) and the tangent to the most superior points at the anterior and posterior edges of the medial tibial plateau (yellow line). The red arrow indicates the evaluated angle.

**Figure 2 jcm-13-00683-f002:**
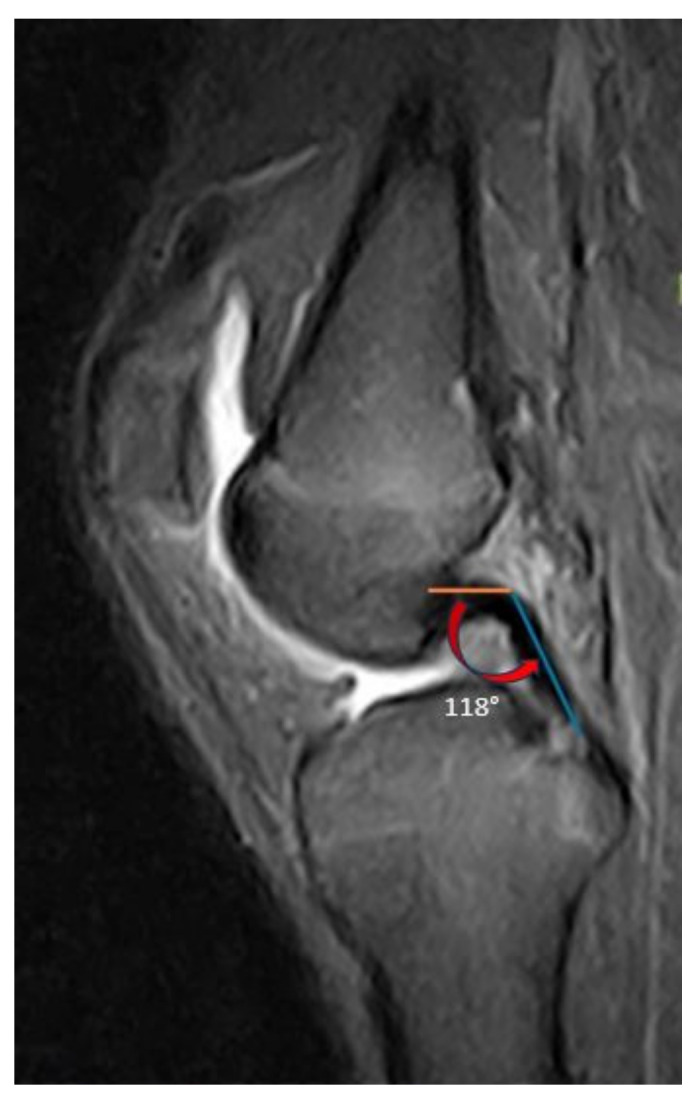
The PCL buckling angle has been described by Yoon et al. [25]. It is formed by the intersection of two lines which follow the proximal and distal portions of the PCL. The red arrow indicates the evaluated angle.

**Table 1 jcm-13-00683-t001:** Baseline characteristics of included patients.

Patients (No. = 96)	Mean ± SD (Range) or No. (%)
Gender	
Male	84 (87.5%)
Female	12 (12.5%)
Age at surgery (years)	28.6 ± 7.8 (18–49)
Dominant limb	71 (74%)
Injury to surgery interval (months)	9.9 ± 13.7 (0.8–60)
Lachman test	
Grade 1	11 (11.4%)
Grade 2	76 (79.2%)
Grade 3	9 (9.4%)
Rotatory instability	
Grade 1	30 (31.2%)
Grade 2	59 (61.5%)
Grade 3	7 (7.3%)
Posterior tibial slope	6.7 ± 1.8 (2.5–10.4)
PCL buckling angle	125 ± 12.9 (97–154)
Concomitant meniscal lesion	
Anterior horn medial meniscus tear	15 (15.6%)
Ramp tear	33 (34.4%)
Lateral meniscus tear	18 (18.8%)
Follow-up (months)	51.6 ± 14 (40–90)

PCL, posterior cruciate ligament; SD, Standard Deviation; No. Number of cases.

**Table 2 jcm-13-00683-t002:** Differences between the isolated ACL and ACL with concomitant ramp lesion groups.

	ACL Group	ACL and Ramp Lesion Group	*p*-Value	95% CI	SED
	Mean ± SD (Range) or No. (%)	Mean ± SD (Range) or No. (%)			
Gender					
Male	51 (81%)	33 (100%)	0.007		
Age at surgery (years)	28.4 ± 8.6 (18–49)	29 ± 6 (19–47)	0.722	−3.933 to 2.733	1.679
Dominant limb	48 (76.2%)	23 (69.7%)	0.625		
Injury to surgery interval (months)	11.1 ± 14.6 (1–60)	7.7 ± 12 (0.8–60)	0.255	−2.475 to 9.275	2.959
Lachman test					
Grade 1	10 (15.9%)	1 (3%)	0.091		
Grade 2	49 (77.7%)	27 (81.8%)	0.793		
Grade 3	4 (6.3%)	5 (15.2%)	0.267		
Rotatory instability					
Grade 1	21 (33.3%)	9 (27.3%)	0.645		
Grade 2	38 (60.3%)	21 (63.6%)	0.479		
Grade 3	4 (6.4%)	3 (9.1%)	0.689		
Posterior tibial slope	6.5 ± 1.7 (3.7–9.7)	7 ± 1.9 (2.5–10.4)	0.192	−1.255 to 0.255	0.380
PCL buckling angle	125.7 ± 12.4 (97–154)	123.8 ± 13.8 (97.5–144)	0.495	−3.601 to 7.401	2.771
Concomitant meniscal tears					
Anterior horn medial meniscus	12 (19%)	3 (9.1%)	0.249		
Lateral meniscus	11 (17.5%)	7 (21.2%)	0.784		
Follow-up (months)	45.9 ± 9.8 (40–90)	51.6 ± 14 (40–90)	0.022	−10.566 to −0.834	2.451

ACL, anterior cruciate ligament; PCL, posterior cruciate ligament; SD, Standard Deviation; No. Number of case; CI, confidence interval; SED, standard error of difference.

## Data Availability

Data are contained within the article.

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
