# Peer review of "A High-Grade Lachman’s Exam Predicts a Ramp Tear of the Medial Meniscus in Patients with Anterior Cruciate Ligament Tear: A Prospective Clinical and Radiological Evaluation"

_jcm, 2024, doi:10.3390/jcm13030683_

Round 1

Reviewer 1 Report

Comments and Suggestions for Authors

The study compares isolated and associated injury ACLRs using imagistic measurements, clinical exams, and demographics. The title and abstract are consistent with the work.

The introduction offers some information on the topic and formulates the objective,

The methodology is presented in a structured way. The subsections should be numbered. A flow chart with the studied population and inclusion/exclusion criteria can be provided. Figure 1 and 2 should be divided and the explanation should be added under each image. The p-values should be written in uppercase, or lowercase please decide, but not both (see line 169 and 177 for e.g.). When describing the statistical steps please start with the normality check test, as it shows you how to present continuous variables: mean±SD or median(IQR).

The results are clear but can be better described in words.

The discussions are very short; in general, the article is short. At least this section should be extended by discussion/making comparison with more complex injuries of the knee (like the unhappy triad for e.g. 10.5312/wjo.v14.i5.268) and presenting rehabilitation perspectives after these injuries (https://doi.org/10.1016/j.promfg.2018.03.122). Limitations of the study are provided.

Overall, the English needs to be refined as there are a lot of repetitions (for e.g. paragraph from line 56 – first 3 sentences start with “various studies”; “previous studies”; “a previous study”).

The references are adequate and properly edited and can be extended as suggested above.

Author Response

Reviewer 1

The study compares isolated and associated injury ACLRs using imagistic measurements, clinical exams, and demographics. The title and abstract are consistent with the work.

The introduction offers some information on the topic and formulates the objective,

The methodology is presented in a structured way. The subsections should be numbered. A flow chart with the studied population and inclusion/exclusion criteria can be provided. Figure 1 and 2 should be divided and the explanation should be added under each image. The p-values should be written in uppercase, or lowercase please decide, but not both (see line 169 and 177 for e.g.). When describing the statistical steps please start with the normality check test, as it shows you how to present continuous variables: mean±SD or median(IQR).

Author Response: Thank you for this comment. We have corrected the test to only include the p values reported with a lowercase p. Additionally, when reporting means we have also reported the standard deviation.

Line Changes: All these changes can be seen in the results section of the paper.

The results are clear but can be better described in words.

Author Response: Thank you for these comments. The results section has been reworked to include more detail and report on more p values.

Line Changes: All these changes can be seen in the results section of the paper.

The discussions are very short; in general, the article is short. At least this section should be extended by discussion/making comparison with more complex injuries of the knee (like the unhappy triad for e.g. 10.5312/wjo.v14.i5.268) and presenting rehabilitation perspectives after these injuries (https://doi.org/10.1016/j.promfg.2018.03.122). Limitations of the study are provided.

Author Response: Thank you for these comments. The discussion section has been totally reworked to include more detail. We have discussed in more detail the complex presentation of ACL tears with other concomitant injuries. We decided not to report on rehabilitation perspectives as we did not find it relevant to the overall topic of the paper, however, we reported on the clinical success of performing a medial meniscus ramp repair.

Line Changes: Discussion section.

Overall, the English needs to be refined as there are a lot of repetitions (for e.g. paragraph from line 56 – first 3 sentences start with “various studies”; “previous studies”; “a previous study”).

Author Response: Thank you for this comment. The introduction has been reworked to attempt to avoid repetitions like this one.

Line Changes: Introduction section, various part throughout.

The references are adequate and properly edited and can be extended as suggested above.

Author Response: Thank you for this comment. The references section has been extended.

Line Changes: An additional 8 references are presented in the paper and the reference section.

Reviewer 2 Report

Comments and Suggestions for Authors

This study entitled “A high-grade Lachman’s exam predicts a ramp tear of the medial meniscus in patients with anterior cruciate ligament tear: a prospective clinical and radiological evaluation” seems to have been generally well executed and written. Furthermore, I believe that this paper will be of great interest to the readers. However, I have only a few suggestions to further improve the quality of this paper.

Abstract

Results

Please state the statistical values of significant findings in your Results.

Keywords

Consider some additional MeSH keywords to readers easier identify your research. 

2.Materials and Methods

Please add the number of the Ethical approval and the date when the approval was gained. Did you register your study (e.g., ClinicalTrials.gov). If yes, please state the date and the number of registration.

4.Discussion

Discussion section is too short. Please expand the Discussion section with the recent findings of similar studies.

Author Response

Reviewer 2

This study entitled “A high-grade Lachman’s exam predicts a ramp tear of the medial meniscus in patients with anterior cruciate ligament tear: a prospective clinical and radiological evaluation” seems to have been generally well executed and written. Furthermore, I believe that this paper will be of great interest to the readers. However, I have only a few suggestions to further improve the quality of this paper.

Abstract

Results

Please state the statistical values of significant findings in your Results.

Author Response: Thank you for this comment. We have addressed this in the results section.

Line Changes: 215-238 A concomitant ACL tear and medial meniscus ramp tear were present in 33 (34.4%) patients. Overall, a concomitant tear of the anterior horn of the medial meniscus (outside the zone of a meniscus ramp tear area) and of the lateral meniscus was present in 15 (15.6%) and 18 (18.8%) patients, respectively.

The cohorts of patients with isolated ACL tears and ACL tears with concomitant medial meniscus ramp tear were separated. All demographic and radiologic factors were compared between groups to assess for associations. A higher rate of male patients was found in the ACL tear with concomitant medial meniscus ramp tear group (p = 0.007). No significant differences were found in terms of age at the time of surgery (p = 0.722), injury to surgery interval (p = 0.255), rotatory instability (Grade 1, p = 0.645; Grade 2, p = 0.479; Grade 3, p = 0.689), posterior tibial slope (p = 0.192), PCL buckling angle (p = 0.495) or concomitant meniscal injuries (Anterior horn medial meniscus, p = 0.249; Lateral meniscus, p = 0.784) between the groups.

Cohen’s kappa coefficients for intraobserver and interobserver reliability of radiological assessments were 0.88 and 0.86 for MRI, and 0.86 and 0.87 for radiographs.

In the univariate regression analysis, a higher grade of the Lachman test was associated with the diagnosis of ACL tear with a concomitant medial meniscus ramp tear (p = 0.027, β = 0.216). Male sex was associated with the diagnosis of ACL tear with a concomitant medial meniscus ramp tear (p = 0.007, β = - 0.191).

The multivariate analysis confirmed the association between the diagnosis of an ACL tear with a concomitant medial meniscus ramp tear with male sex (p = 0.013, β = - 0.359) and a higher grade of the Lachman test (p = 0.049, β = 0.203).

Keywords

Consider some additional MeSH keywords to readers easier identify your research. 

Author Response: Thank you for this comment. Additional keywords were added.

Line Changes: 30-31 Keywords: Anterior cruciate ligament, Medial meniscus ramp, Lachman’s exam, Posterior tibial slope, Rotatory instability, Clinical diagnostics

2.Materials and Methods

Please add the number of the Ethical approval and the date when the approval was gained. Did you register your study (e.g., ClinicalTrials.gov). If yes, please state the date and the number of registration.

Author Response: Thank you for this comment. This section was added.

Line Changes:

4.Discussion

Discussion section is too short. Please expand the Discussion section with the recent findings of similar studies

Author Response: Thank you for this comment. The entire discussion section was reworked.

Line Changes: Discussion section.
